# Integrating Activity-Guided Strategy and Fingerprint Analysis to Target Potent Cytotoxic Brefeldin A from a Fungal Library of the Medicinal Mangrove *Acanthus ilicifolius*

**DOI:** 10.3390/md20070432

**Published:** 2022-06-29

**Authors:** Cui-Fang Wang, Jie Ma, Qian-Qian Jing, Xi-Zhen Cao, Lu Chen, Rong Chao, Ji-Yong Zheng, Chang-Lun Shao, Xiao-Xi He, Mei-Yan Wei

**Affiliations:** 1Key Laboratory of Marine Drugs, The Ministry of Education of China, School of Medicine and Pharmacy, College of Food Science and Engineering, Ocean University of China, Qingdao 266003, China; wangcuifang0115@163.com (C.-F.W.); majie346@163.com (J.M.); jingqianqian1231@163.com (Q.-Q.J.); caoxizhen2022@163.com (X.-Z.C.); cl2014049048@163.com (L.C.); chaorong1122@163.com (R.C.); 2State Key Laboratory for Marine Corrosion and Protection, Luoyang Ship Material Research Institute (LSMRI), Qingdao 266061, China; zhengjy@sunrui.net

**Keywords:** *Acanthus ilicifolius*, endophytic fungi, fungal metabolites, cytotoxic activity, activity-guided strategy

## Abstract

Mangrove-associated fungi are rich sources of novel and bioactive compounds. A total of 102 fungal strains were isolated from the medicinal mangrove *Acanthus ilicifolius* collected from the South China Sea. Eighty-four independent culturable isolates were identified using a combination of morphological characteristics and internal transcribed spacer (ITS) sequence analyses, of which thirty-seven strains were selected for phylogenetic analysis. The identified fungi belonged to 22 genera within seven taxonomic orders of one phyla, of which four genera *Verticillium*, *Neocosmospora*, *Valsa*, and *Pyrenochaeta* were first isolated from mangroves. The cytotoxic activity of organic extracts from 55 identified fungi was evaluated against human lung cancer cell lines (A-549), human cervical carcinoma cell lines (HeLa), human hepatoma cells (HepG2), and human acute lymphoblastic leukemia cell lines (Jurkat). The crude extracts of 31 fungi (56.4%) displayed strong cytotoxicity at the concentration of 50 μg/mL. Furthermore, the fungus *Penicillium* sp. (HS-N-27) still showed strong cytotoxic activity at the concentration of 25 µg/mL. Integrating cytotoxic activity-guided strategy and fingerprint analysis, a well-known natural Golgi-disruptor and Arf-GEFs inhibitor, brefeldin A, was isolated from the target active strain HS-N-27. It displayed potential activity against A549, HeLa and HepG2 cell lines with the IC_50_ values of 101.2, 171.9 and 239.1 nM, respectively. Therefore, combining activity-guided strategy with fingerprint analysis as a discovery tool will be implemented as a systematic strategy for quick discovery of active compounds.

## 1. Introduction

Cancer stands in the frontline among leading killers worldwide and the annual mortality rate is expected to reach 16.4 million by 2040 [1,2]. The marine environment has the potential to produce candidate compounds (structures) as leads to drugs, or actual drugs, as has been actively discussed for the last 50 or so years [3,4,5]. Nowadays, several compounds have led to drugs, especially in the area of cancer, such as trabectedin, and eribulin, which were discovered under the cytotoxic activity-guided approach [3,4,5,6]. Brefeldin A (BFA), a well-known natural Golgi-disruptor and Arf-GEFs inhibitor, was first isolated from *Penicillium decumbens* in 1958 [7,8] and subsequently identified only from the marine-derived genus *Penicillium* [9]. Previous studies reported that BFA showed strong anticancer activity in a variety of cancers, including colorectal, prostate, lung, and breast cancers [10,11]. BFA is considered as a promising leading molecule for developing anticancer drugs. 

The mangrove forests are a complex ecosystem growing in tropical and subtropical intertidal estuarine zones and nourish a diverse group of microorganisms [12,13]. Microorganisms associated with mangrove environments are a major source of antimicrobial agents and also produce a wide range of important medicinal compounds, including enzymes, antitumor agents, insecticides, vitamins, immunosuppressants, and immune modulators [13,14,15,16,17]. Among the mangrove microbial community, mangrove associated fungi were the second-largest ecological group of the marine fungi [13,14]. Up to December 2020, at least 1387 new structures have been isolated and identified from a diverse range of mangrove-derived fungi (325 strains), which belong to about 69 genera. Furthermore, about 40.7% (530) of the 1300 new compounds displayed a wide range of pharmacological activities, and the antitumor (mainly cytotoxicity) function is noteworthy and visible, accounting for 34% (196 compounds) of the active compounds. Therefore, mangrove associated fungi are a rich source of structurally unique and diverse bioactive secondary metabolites [13]. 

*Acanthus ilicifolius* is widely distributed from India to southern China, tropical Australia and the Western Pacific islands, throughout Southeast Asia [18]. Various classes of bioactive compounds including alkaloids, benzoxazinoids, lignans, flavanoids, triterpenoids and steroids have been obtained from *A. ilicifolius* [18,19,20]. In addition, up to December 2020, a total of 22 strains belonging to 9 genera have been reported, which produced 95 new secondary metabolites. The endophytic fungi derived from *A. ilicifolius* are one of the most favored to be studied [13], yet little attention has been paid to the fungal communities associated with *A. ilicifolius*. 

Investigating new bioactive natural products from marine fungi is a major and constant research focus in our laboratory [21,22,23,24]. Natural product researchers also face the challenge of targeting the discovery of bioactive compounds from a microbial resource library. The present work aims to integrate activity-guided strategy and fingerprint analysis to target the potent cytotoxic compounds from a fungal library of the medicinal mangrove *A. ilicifolius* (Figure 1). The cultivable fungi associated with the medicinal mangrove *A. ilicifolius* from the South China Sea were firstly systematic evaluated for their diversity. Furthermore, integrating the cytotoxic activity-guided strategy, the target active strains were quickly identified. Combined with fingerprint analysis, a potent cytotoxic activity compound, brefeldin A, was isolated from the target active strains. The combination of activity-guided strategy and fingerprint analysis could improve the efficiency of discovering active compounds in crude extracts from a complex and diverse fungal library.

## 2. Results

### 2.1. Cultivable Fungi’s Phylogeny and Diversity

A total of 102 fungal isolates were obtained from *Acanthus ilicifolius* using the PDA medium with four salt gradients of 3%, 5%, 7% and 10%. Duplicated strains were removed using a detailed morphological approach. Consequently, eighty-four independent strains were selected for sequencing and identification based on ITS sequences. According to the sequences deposited into NCBI, the 84 strains belonged to the phylum Ascomycota including seven taxonomic orders: *Hypocreales*, *Xylariales*, *Diaporthales*, *Eurotiales*, *Pleosporales*, *Capnodiales*, *Botryosphaeriaceae* and 22 genera: *Trichoderma*, *Hypocrea, Acremonium*, *Verticillium*, *Fusarium*, *Neocosmospora*, *Pestalotiopsis*, *Diaporthe*, *Phomopsis*, *Valsa, Colletotrichum, Penicillium, Eupenicillium*, *Aspergillus, Talaromyces*, *Pyrenochaeta*, *Pleosporales*, *Curvularia*, *Alternaria*, *Cladosporium*, *Phyllosticta*, and *Lasiodiplodia* (Table 1). These identified fungi and their best matches in the NCBI database are summarized in Appendix A. Most of the isolates matched their closest relatives with 98 to 100% similarity, except for HS-G-02 (97%) and HS-G-06 (95%), which indicated that they were new species. Both of the fungi HS-G-06 and HS-G-02 further enriched the diversity of mangrove fungi. Further analysis of the isolated fungi showed that *Eurotiales* was the dominant group with identified fungi, followed by *Hypocreales*. The fungal community was dominated by *Penicillium*, comprising 21 isolates, followed by *Fusarium*, *Aspergillus*, and *Eupenicillium* with 15, 14, and 10 isolates, respectively. Some of the genera, such as *Trichoderma, Phomopsis* and *Cladosporium* obtained six, five and five, respectively. Most of the remaining genera occurred as singletons or doubletons. 

In addition, the species of fungi isolated from different parts of *A. ilicifolius* were quite different (Figure 2). The results showed that some genera of fungi were isolated only from one part. For example, *Phomopsis* and *Acremonium* were isolated only from the stem. *Colletotrichum*, *Curvularia*, and *Alternaria* were isolated only from the leaf. *Valsa*, *Hypocrea*, and *Neocosmospora* were isolated only from the soil. *Diaporthe*, *Talaromyces*, and *Pyrenochaeta* were isolated only from the leaf.

Further phylogenetic analysis was carried out on 37 strains. These 37 independent individuals were selected as the representative strains because they belong to different fungal species after we aligned the sequences with the BioEdit software (Appendix A). The phylogenetic tree of fungi in the order *Hypocreales* based on ITS gene sequence is presented in Appendix A. Furthermore, the fingerprints of secondary metabolites of fungi from different species and genera were analyzed (Appendix A).

### 2.2. The Cytotoxicity of Cultivable Fungal Extracts

The organic extracts of 55 identified fungi were evaluated for their cytotoxic activities against human lung cancer cell line (A-549), human cervical carcinoma cell (HeLa), human hepatoma cells (HepG2) and Jurkat tumour cell lines at the concentration of 50 μg/mL (Figure 3a). To identify active strains for further research as potential cytotoxic strains, the relative inhibition rate of A-549, HeLa and HepG2 cell lines should greater than 70%, and the relative inhibition rate of Jurkat cell line should greater than 60%. The results showed that these fungi showed different inhibition rates to different cell lines. The number of the fungi showing activity against A-549, HeLa, HepG2, and Jurkat tumour cell lines were 17, 17, 19 and 24, respectively (Appendix A). The crude extracts of 31 fungi displayed cytotoxicity against the test cell lines, of which 21 fungi showed selective inhibitory activity on different tested cell lines,; for example, *Fusarium* sp. showed selective inhibitory activity on HeLa cell lines. Most fungi showed strong selective inhibitory activity on Jurkat cell lines. Interestingly, the remaining 10 fungi belonging to the two orders *Eurotiales* and *Hypocreales*, displayed a broad-spectrum strong cytotoxic activity, such as *Penicillium* sp. (HS-N-23, HS-N-27, HS-N-29, and HS-G-01), *Eupenicillium* sp. (HS-N-25), *Trichoderma* sp. (HS-01 and HS-N-04), *Aspergillus* sp. (HS-G-04 and HS-Y-27), and *Verticillium* sp. (HS-N-28).

The crude extracts were further reduced in concentration for the activity test. The results showed that only the two active strains of *Penicillium* sp. (HS-N-27 and HS-N-29) still showed strong inhibitory activity against all the texted cell lines at the concentration of 25 μg/mL. Cytotoxic metabolites were isolated from the endophytic fungus *Penicillium chermesinum*, leading to the discovery of a cysteine-targeted Michael acceptor as a pharmacophore for fragment-based drug discovery, bioconjugation and click reactions [25]. The heteroatom-containing new compounds 2-hydroxyl-3-pyrenocine-thio propanoic acid and 5,5-dichloro-1-(3,5-dimethoxyphenyl)-1,4-dihydroxypentan-2-one, which were isolated from a deep-sea *Penicillum citreonigrum* XT20-134, showed potent cytotoxicity to the human hepatoma tumor cell Bel7402 [26]. Additionally, the active strains HS-N-28, HS-G-01, and HS-N-25 showed strong selective inhibitory activity against A-549, HeLa and HepG2 cell lines. The active strains HS-Y-27, HS-N-23, HS-G-04, and HS-N-28 showed strong selective inhibitory activity against HeLa cell lines. The active strains HS-G-01, and HS-N-25 showed strong selective inhibitory activity against HepG2 and A549 cell lines. Obviously, these active strains are important microbial resources and have the potential for interesting cytotoxic compounds (Figure 3b). 

### 2.3. Isolation and Identifcation of Compounds **1**–**7**

As the two active strains *Penicillium* sp. (HS-N-27 and HS-N-29) showed strong cytotoxic activity against all the tested cell lines at the concentration of 25 μg/mL, both of the *Penicillium* sp. fungi were selected as the target strains. Combining cytotoxic activity-guided strategy with fingerprint analysis, compound **1** was obtained from the fermentation broth of the two active strains HS-N-27 and HS-N-29. By comparison of NMR data with the reported literature, the structure was identified as brefeldin A (Figure 5), which was a 13-membered macrolactone with a cyclopentane substituent [7]. BFA is a well-known natural Golgi-disruptor and Arf-GEFs inhibitor [8]. Combining morphological characteristics and fingerprint analysis of metabolites (Appendix A), the two fungi HS-N-27 and HS-N-29 were identified as different individuals of the same *Penicillium* sp. species. The neighbor-joining of the phylogenetic tree of the target active strain *Penicillium* sp. (HS-N-27) in *Hypocreales* order fungi from *A. ilicifolius* based on ITS sequences is shown in Figure 4.

The genus *Aspergillus* is one of the dominant producers of new natural products [13]. The fingerprint analysis showed that the metabolites of *Aspergillus flavus* (HS-N-06) were relatively single and that *A. candidus* (HS-Y-23) was rich in metabolites with strong special UV absorption peak (Appendix A). The secondary metabolites of the two fungal strains were further studied. Under the guidance of chemical technology, 5-hydroxymethylfuran-3-carboxylic acid (**2**) was obtained from the fermentation broth of *A. flavus* (HS-N-06) [27]. Terphenyllin (**3**) was obtained from the fermentation broth of *A. candidus* (HS-Y-23), which showed weak cytotoxic activity against HeLa cell lines with the IC_50_ value of 19.0 µM [28]. In addition, 5-hydroxy-3-hydroxymethyl-2-methyl-7-methoxychromone (**4**), indolyl-3-carboxylic acid (**5**), and trichodermamides A (**6**) and D (**7**) were obtained from the fermentation broth of *Trichoderma harzianum* (HS-N-04) [29,30,31]. The structures of isolated and identified compounds were in Figure 5.

## 3. Discussion

Mangrove-associated fungi are rich in diversity and can produce impressive quantities of metabolites with promising biological activities that may be useful to humans as novel physiological agents [13,14,15,16,17]. The phylogenetic diversity of culturable fungi derived mangrove species *Rhizophora stylosa* and *R. mucronata* collected from the South China Sea has been reported [32]. The endophytic fungi derived from *A. ilicifolius* areamong the most favored to be studied. Up to December 2020, only 22 strains associated with *A. ilicifolius* belonging to 9 genera have been reported [13]. Investigation on phylogenetic diversity of *A. ilicifolius* associated fungi is relatively rare. In this study, 84 of the 102 isolates were successfully classified at the genus level based on ITS sequences with relatives in the NCBI database (Appendix A). The identified fungi belonged to 22 genera, of which four genera *Verticillium*, *Neocosmospora*, *Valsa*, and *Pyrenochaeta* were first isolated from mangroves. (Appendix A). Two strains HS-G-02 (97%) and HS-G-06 (95%) with low similarity indicated that they should be new species, which further enriched the diversity of mangrove fungi. The new strains may produce a variety of commercially interesting and potentially useful products. The above results indicated that a high diversity of fungi can be recovered from *A. ilicifolius* in the South China Sea.

Further analysis of the isolated fungi showed that *Eurotiales* was the dominant group with identified fungi accounted for 45.1%, followed by *Hypocreales*. The fungal community comprising *Penicillium* accounted for 20.6%, followed by *Fusarium*, and *Aspergillus*. It was reported that *Penicillium* (283, 20%), *Aspergillus* (246, 18%), and *Pestalotiopsis* (88, 6%) are the dominant producers of new natural products (1384) isolated from mangrove-associated fungi, comprising more than 45% of the total molecules [13]. The fungi obtained from *A. ilicifolius* could provide abundant microbial resources for the discovery of new compounds.

Natural product researchers face the challenge of maximizing the discovery of new or potent compounds from a microbial resource library. Combining activity-guided strategy with fingerprint analysis as a discovery tool will be implemented as a systematic strategy for quick discovery of active compounds. The crude extracts of 56.4% fungi displayed strong cytotoxicity. Interestingly, the remaining 10 fungi belonging to the two orders *Eurotiales* and *Hypocreales*, displayed a broad-spectrum strong cytotoxic activity. Furthermore, integrating cytotoxic activity-guided strategy and fingerprint analysis, a strong cytotoxic active compound brefeldin A was isolated from the target active strain HS-N-27. Brefeldin A is a well-known natural Golgi-disruptor and Arf-GEFs inhibitor, and shows strong anticancer activity in a variety of cancers [8,9,10,11]. BFA is considered as a promising leading molecule for developing anticancer drugs. As the metabolites of the fungi *Penicillium* sp. (HS-N-27) are relatively simple and BFA is easily separated and purified, this provides the source of compounds for the study of the medicinal properties of BFA. A series of BFA derivatives with antileukemia activity had been reported in terms of the semi-synthesis, cytotoxic evaluation, and structure-activity relationships [9]. This method, combining activity-guided strategy with fingerprint analysis, could improve the efficiency of discovering active compounds.

## 4. Materials and Methods

### 4.1. Sampling Site and Plant Material

The medicinal mangrove *A. ilicifolius*, which was authenticated by Prof. Fengqin Zhou (Shandong University of Traditional Chinese Medicine) was collected from the South China Sea. The samples were stored at the Key Laboratory of Marine Drugs, the Ministry of Education of China, School of Medicine and Pharmacy, Ocean University of China, Qingdao, China.

### 4.2. Isolation of Cultivable Fungi

To obtain the fungi associated with medicinal mangrove *A. ilicifolius* within different parts of the plant, the surface sterilization of each part from *A. ilicifolius* was carried out following an isolation as Qin et al. described with some modifications [33]. The root, stem and leaf of *A. ilicifolius* samples were washed with sterile artificial seawater for three times to remove the microorganisms and sediment attached to the surface. Appropriate samples were taken, using scissors or scalpel to cut all parts, including root, stem and leaf, with attention to the integrity of sampling. Then, the sample was soaked in 75% alcohol for 30 s, and the water on the sample was sucked up with sterile filter paper. The sample was cut into 1 cm³ pieces for fungal isolation.

The methods of tissue sectioning and tissue homogenization were used to isolate fungi. Tissue sectioning method: The tissues of 1 cm³ pieces were inoculated into PDA medium (200.0 g of potato extract, 20.0 g of glucose in 1 L of seawater with four salinities of 3%, 5%, 7% and 10% respectively) in a sterile environment. In order to improve the utilization of the plate and to separate more microorganisms, the medium plate was generally divided into three areas, and 2–3 pieces of tissue were placed in each area of the PDA medium with four salt gradients of 3%, 5%, 7% and 10%. Tissue homogenization method: The tissue was ground in 2 mL of sterile artificial seawater with a mortar in a sterile environment, and then the resulting homogenate was diluted with sterile artificial seawater at three dilutions (1:10, 1:100, and 1:1000). 100 μL of each dilution was plated in quadruplicate onto corresponding medium for fungal cultivation. The inoculated plates were cultured at 25 °C for 2 days. The fungi were replated onto new PDA plates several times until the morphology of the fungi could be distinguished. The obtained fungal strains were deposited at the Key Laboratory of Marine Drugs, the Ministry of Education of China, School of Medicine and Pharmacy, Ocean University of China, Qingdao, China.

### 4.3. Genomic DNA Extraction, PCR Amplifcation, Sequencing and Phylogenetic Analysis

The genomic DNA extraction was conducted using the Fungal DNA kits (E.Z.N.A., Omega, Norcross, GA, USA) according to the manufacturer’s protocol. The internal transcribed spacer (ITS1-5.8S-ITS2) regions of the fungi were amplified with the universal ITS primers, ITS1F (5′-CTTGGTCATTTAGAGGAAGTAA-3′) and ITS4 (5′-TCCTCCGCTTATTGATATGC-3′) using the polymerase chain reaction (PCR) [34]. The PCR was performed through the following cycle: initial denaturation at 94 °C for 5 min, 30 cycles of 94 °C denaturation for 40 s min, 52 °C annealing for 40 s, 72 °C extension for 1 min; with a final extension at 72 °C for 10 min. Finally, the amplified products were submitted for sequencing (Invitrogen, Shanghai, China) and a BLASTN search was used to search for sequences of the closest match in the GenBank by Basic Local Alignment Search Tool (BLAST) programs database. 

The sequences of fungal ITS regions obtained from *A. ilicifolius* were compared with the related sequences in the National Center for Biotechnology Information (NCBI). Each of these sequences was then aligned to sequences available in the NCBI database to determine the identity of the sequence, which further determined the species and genera of fungi. All fungal ITS sequences were aligned using the BioEdit software, applying the default parameters. The phylogenetic tree was generated using neighbor-joining (NJ) algorithms in the MEGA 7 software (version 7.0, Mega Limited) combined with bootstrap analysis using 1000 replicates incorporating fungal sequences showing the highest homology to sequences amplified.

### 4.4. General Experimental Procedures

The Agilent DD2 NMR spectrometer (JEOL, Tokyo, Japan) at 500 MHz and 125 MHz frequency was used for ^1^H and ^13^C NMR spectra respectively. The vacuum column chromatography silica gel (200–300 mesh, Qing Dao Hai Yang Chemical Group Co, Qingdao, China), silica gel plates for thin layer chromatography (G60, F-254, and Yan Tai Zi Fu Chemical Group Co, Yan Tai, China), and reverse phase octadecylsilyl silica gel column were used for the separation of compounds. UPLCMS spectra were measured on Waters UPLC® system (Waters Ltd., Milford, MA, USA) using a C_18_ column (ACQUITY UPLC® BEH C_18_, 2.1 × 50 mm, 1.7 μm; 0.5 mL/min) and ACQUITY QDA ESIMS scan from 150 to 1000 Da was used for the analysis of fungal extracts and ESI-MS spectra of the compounds. Semipreparative HPLC was performed on a Hitachi L-2000 system (Hi-tachi Ltd., Tokyo, Japan) using a C_18_ column (Kromasil 250 × 10 mm, 5 μm, 2.0 mL/min). 

### 4.5. Fungal Fermentation and Chemical Extraction

The 55 fungal isolates were fermented in a 500 mL conical flask containing 250 mL PDA liquid medium. The fungi were shaken at 28 °C, 120 rpm for 7 days. Each exper-iment was conducted in three parallels. The fermentation broth was extracted three times with an equal volume of EtOAc and the whole EtOAc solutions were evaporated under reduced pressure to give the dried extracts.

### 4.6. Cytotoxic Assay

The cytotoxic activity was evaluated against human lung cancer cell line (A-549), human cervical carcinoma cells (HeLa), human hepatoma cells (HepG2) and Jurkat tumor cell lines by the MTT method, with adriamycin as a positive control [35]. The organic extracts, and adriamycin (the positive control) were dissolved in DMSO with the concentration of 50 µg/mL, 25 µg/mL, and 1 µM, respectively for bioassay.

### 4.7. Extraction and Isolation of Compounds

The organic extract of *Penicillium* sp. (HS-N-27) showed strong cytotoxic activity. The organic extract of the *Penicillium* sp. (HS-N-27) was subjected to silica gel column chromatography (CC) and eluted by a gradient of petroleum ether (PE)/ethyl acetate (EA) and then EA/MeOH to generate nine fractions (Fr. 1–9). All the fractions were further evaluated for cytotoxic activity. The Fr. 5 showed strong cytotoxic activity. Combined with fingerprint analysis,-the composition of Fr. 5 is relatively simple (Appendix A); it was further purified by semipreparative HPLC (MeOH−H_2_O, 80%; 2 mL/min) to obtain **1** (BFA) (19.0 mg).

The organic extract of the *Aspergillus flavus* (HS-N-06) was subjected to silica gel vacuum liquid chromatography (VLC) and eluted by a gradient of PE/EA and then EA/MeOH to afford four subfractions (Fr. 1−Fr. 4). Fr. 3 was separated by ODS CC (MeOH−H_2_O, 30–50%) to afford **2** (17.0 mg).

The organic extract of the *A. aculeatus* (HS-Y-23) was subjected to silica gel vacuum liquid chromatography (VLC) and eluted by a gradient of PE/EA and then EA/MeOH to afford seven subfractions (Fr. 1−Fr. 7). Fr. 4 was separated by ODS CC (MeOH−H_2_O, 30–100%) and then purified by semipreparative HPLC (MeOH−H_2_O, 70%; 2 mL/min) to afford **3** (7.0 mg).

The organic extract of the *Trichoderma harzianum* (HS-N-04) was subjected to silica gel column chromatography (CC) and eluted by a gradient of PE/EA and then EA/MeOH to generate six fractions (Fr. 1–6). Fr. 2 was further purified by using CC to generate five fractions (Fr. 2-1–2-5). Fr.2-2 was separated by normal phase silica gel column chromatography and purified by semi preparative HPLC (MeOH−H_2_O, 85%; 2 mL/min) to obtain **4** (10.0 mg). Fr.2-4 was separated by normal phase silica gel column chromatography and semipreparative HPLC to obtain **5** (2.4 mg). Fr.4 was separated into six fractions by Sephadex LH-20 eluting with MeOH gel column. Fr.4-2 was purified by semipreparative HPLC (MeOH−H_2_O, 70%; 2 mL/min) to yield **6** (5.3 mg). Fr.4-4 was purified by semipreparative HPLC (MeOH−H_2_O, 76%; 2 mL/min) to obtain **7** (5.0 mg). 

## 5. Conclusions

This is the first systematic report on the phylogenetic diversity of fungi from mangrove *A. ilicifolius*. Four genera *Verticillium*, *Neocosmospora*, *Valsa*, and *Pyrenochaeta*, which were first isolated from mangroves, further enriched the diversity of mangrove fungi. Thirty-one strains of fungi displayed strong cytotoxicity to different cell lines, which was the important microbial resource for the discovery of cytotoxic compounds. Furthermore, by integrating cytotoxic activity-guided strategy and fingerprint analysis, a potent cytotoxic activity compound was quickly isolated from target active strains. This method, combining activity-guided strategy with fingerprint analysis, could improve the efficiency of discovering active compounds. 

## Figures and Tables

**Figure 1 marinedrugs-20-00432-f001:**
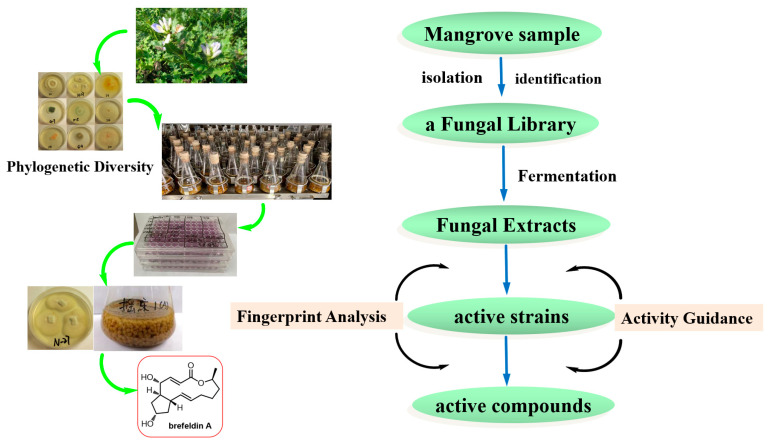
The detailed flowchart of this study.

**Figure 2 marinedrugs-20-00432-f002:**
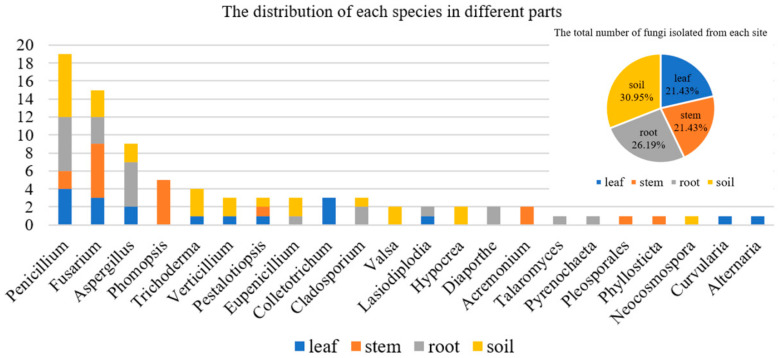
The distribution of each species in different parts.

**Figure 3 marinedrugs-20-00432-f003:**
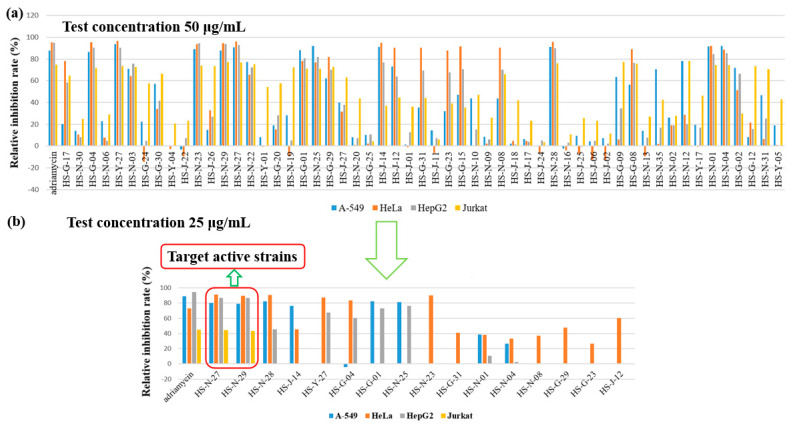
Cytotoxicity of organic extracts of 55 identified fungi. (**a**) The cytotoxic activities of 55 identified fungi against A-549, HeLa, HepG2 and Jurkat tumour cell lines at the concentration of 50 μg/mL. (**b**) The cytotoxic activities of the active strains at the concentration of 25 μg/mL.

**Figure 4 marinedrugs-20-00432-f004:**
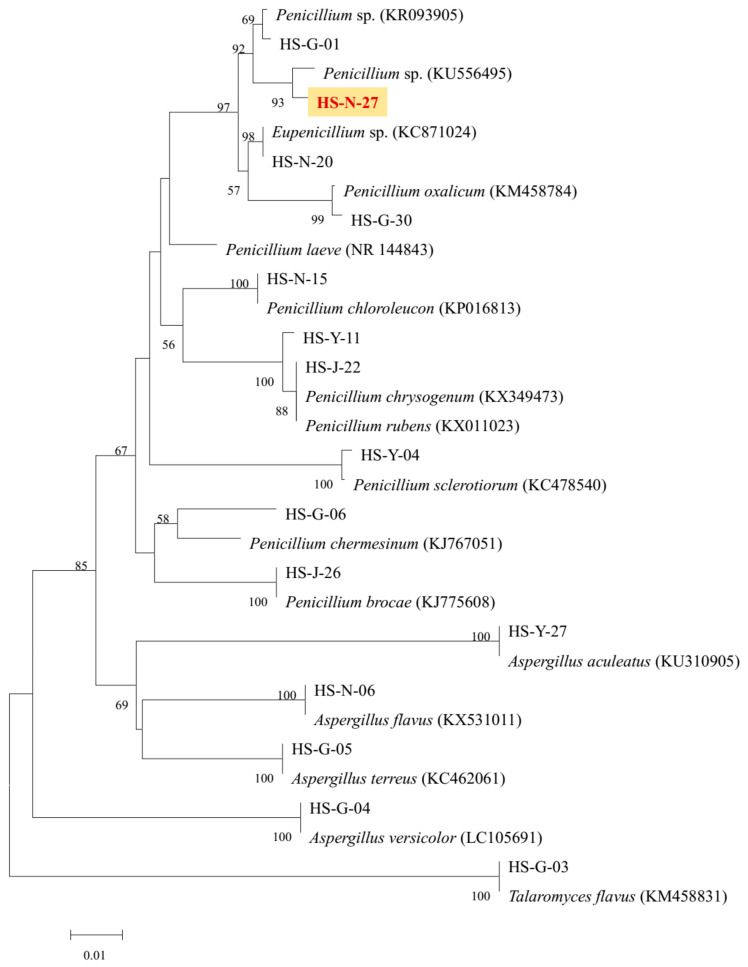
The neighbor-joining of phylogenetic tree of HS-N-27 fungi in *Hypocreales* order fungi. The values at each node represent the bootstrap values from 1000 replicates, and the scale bar = 0.01 substitutions per nucleotide.

**Figure 5 marinedrugs-20-00432-f005:**
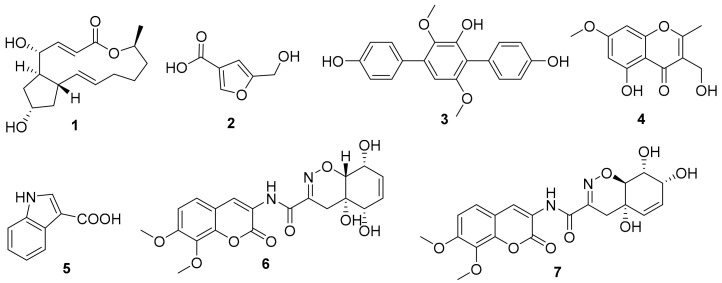
Structures of isolated and identified compounds.

**Table 1 marinedrugs-20-00432-t001:** The classification of cultivable fungi associated with *Acanthus ilicifolius*.

	Phylum	Class	Order	Genus	Number
	Ascomycota	Sordariomycetes	*Hypocreales*	*Trichoderma*	6
				*Hypocrea*	2
				*Acremonium*	2
				*Verticillium*	3
				*Fusarium*	15
				*Neocosmospora*	1
				*Colletotrichum*	3
			*Xylariales*	*Pestalotiopsis*	3
			*Diaporthales*	*Diaporthe*	2
				*Phomopsis*	5
				*Valsa*	2
		Eurotiomycetes	*Eurotiales*	*Penicillium*	21
				*Eupenicillium*	10
				*Aspergillus*	14
				*Talaromyces*	1
		Dothideomycetes	*Pleosporales*	*Pyrenochaeta*	1
				*Pleosporales*	1
				*Curvularia*	1
				*Alternaria*	1
			*Capnodiales*	*Cladosporium*	5
			*Botryosphaeriaceae*	*Phyllosticta*	1
				*Lasiodiplodia*	2
**Total**	**1**	**3**	**7**	**22**	**102**

## Data Availability

Data are contained within the article or Appendix A.

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
