# Peer review of "Integrating Activity-Guided Strategy and Fingerprint Analysis to Target Potent Cytotoxic Brefeldin A from a Fungal Library of the Medicinal Mangrove Acanthus ilicifolius"

_marinedrugs, 2022, doi:10.3390/md20070432_

Round 1

Reviewer 1 Report

What is the novelty?

No analysis fingerprint

The introduction does not say anything about why they decided to look for Brefeldin A and in these mangrove mushrooms, why look for this compound at all

Title does not match content

Aspergillus flavus (HS-N-06), A. candidus (HS-Y-23), Trichoderma harzianum (HS-N-04) why were these strains selected?

Why did you decide that it was Brefeldin A in the extract that was responsible for the significant cytotoxic activity?

What is the core of the work?

Why only epithelial cells were used, and not different types of cell lines, and also normal cell lines were not used?

Author Response

 Responds to Reviewers' Comments

Reviewer #1

1.What is the novelty? 2. No analysis fingerprint. 3. The introduction does not say anything about why they decided to look for Brefeldin A and in these mangrove mushrooms, why look for this compound at all. 4. Title does not match content. 5. Aspergillus flavus (HS-N-06), A. candidus (HS-Y-23), Trichoderma harzianum (HS-N-04) why were these strains selected? 6. Why did you decide that it was Brefeldin A in the extract that was responsible for the significant cytotoxic activity? 7. What is the core of the work? 8. Why only epithelial cells were used, and not different types of cell lines, and also normal cell lines were not used?

Response:

Thanks for your suggestion. As we know, this is the first report of systematically phylogenetic diversity of fungi from mangrove A. ilicifolius. Furthermore, the fingerprints of secondary metabolites of fungi from different species and genera were analyzed and the cytotoxic activity of fungal metabolites was evaluated. Integrating the cytotoxic activity-guided strategy with fingerprint analysis, a potent cytotoxic activity compound brefeldin A, was quickly isolated from the target active strains. This method that combined activity-guided strategy with fingerprint analysis could improve the efficiency of discovering active compounds in crude extracts from a complex and diverse fungal library.

As the identified fungi were belonged to 22 genera, the fingerprint analysis of the organic extract of fungi from different species and genera were added in Figure S4. The metabolites of some fungi are relatively single, while some fungi are rich in metabolites, and some fungi have special UV absorption peaks. The genus Aspergillus is one of the dominant producers of new natural products. Aspergillus flavus (HS-N-06) had special UV absorption peak in 2.9min, and the metabolites are relatively single. The fungus A. candidus (HS-Y-23) is rich in metabolites with strong special UV absorption peak and the quality of crude extract is higher than others. Trichoderma harzianum (HS-N-04) had special UV absorption peak. The three fungi were selected.

The fungus Penicillium sp. (HS-N-27) still showed strong cytotoxic activity at the concentration of 25 µg/mL. The organic extract of the Penicillium sp. (HS-N-27) was subjected to silica gel column chromatography (CC) and eluted by a gradient of petroleum ether (PE)/ ethyl acetate (EA) and then EA/MeOH to generate nine fractions (Fr. 1−9). All the fractions were further evaluated for cytotoxic activity. The Fr. 5 showed strong cytotoxic activity. Combined with fingerprint analysis, the composition of Fr. 5 is relatively simple (Figure S5), and brefeldin A (BFA) was obtained. BFA displayed potential activity against A549, HeLa and HepG2 cell lines with the IC50 values of 101.2, 171.9 and 239.1 nM, respectively. So BFA in the extract that was responsible for the significant cytotoxic activity. As this method that combined activity-guided strategy with fingerprint analysis was used to improve the efficiency of discovering potent cytotoxic active compounds in crude extracts from a complex and diverse fungal library, the toxicity to normal cells was not considered.

We added the following paragraph in the introduction to explained why we decided to look for Brefeldin A. Cancer stands in the frontline among leading killers worldwide and the annual mortality rate is expected to reach 16.4 million by 2040 [1−2]. The marine environment has the potential to produce candidate compounds (structures) as leads to, or even di-rect drugs from, which has been actively discussed for the last 50 or so years [3−5]. Nowadays, several compounds have led to drugs, especially in the area of cancer, such as trabectedin, and Eribulin, which were discovered under the cytotoxic activity-guided [3−6]. Brefeldin A (BFA), a well-known natural Golgi-disruptor and Arf-GEFs inhibitor, was first isolated from Penicillium decumbens in 1958 [7,8] and subsequently identified only from the marine-derived geneus Penicillium [9]. Previous studies reported that BFA showed obvious anticancer activity in a variety of cancers, including colorectal, prostate, lung, and breast cancers [10,11]. BFA is considered as a promising leading molecule for developing anticancer drugs.

  1. Sung, H.; Ferlay, J.; Siegel, R.L; Laversanne, M.; Soerjomataram, I.; Jemal, A.; Bray, F. Global Cancer Statistics 2020: GLOBOCAN estimates of incidence and mortality worldwide for 36 cancers in 185 countries. CA: Cancer J. Clin. 2021, 71, 209–249. DOI: 10.3322/caac.21660
  2. Mohan, C.D.; Rangappa, S.; Nayak, S.C.; Jadimurthy, R.; Wang, L.; Sethi, G.; Garg, M.; Rangappa, K.S. Bacteria as a treasure house of secondary metabolites with anticancer potential. Semin. Cancer Biol. 2021, https://doi.org/10.1016/j.semcancer.2021.05.006
  3. Newman, D.J.; Cragg, G.M. Drugs and Drug Candidates from Marine Sources:An Assessment of the Current “State of Play”. Planta Med. 2016, 82, 775–789. DOI http://dx.doi.org/10.1055/s-0042-101353
  4. Newman, D.J.; Cragg, G.M. Natural Products as Sources of New Drugs over the Nearly Four Decades from 01/1981 to 09/2019. Nat. Prod. 2020, 83, 770−803. https://dx.doi.org/10.1021/acs.jnatprod.9b01285
  5. Jimenez, P.C.; Wilke, D.V.; Branco, P.C.; Bauermeister, A.; Rezende‐Teixeira, P.; Gaudêncio, S.P.; Costa‐Lotufo, L.V. Enriching cancer pharmacology with drugs of marine origin. J. Pharmacol. 2020, 177, 3–27. https://doi.org/10.1111/bph.14876
  6. Wright, A.E.; Forleo, D.A.; Gunawardana, G.P.; Gunasekera, S.P.; Koehn, F.E.; McConnell, O.J.; Antitumor Tetrahydroisoquinoline Alkaloids from the Colonial Ascidian Ecteinascidia turbinate. Org. Chem., 1990, 55, 4509−4512. https://doi.org/10.1021/jo00302a006
  7. Singleton, V.L.; Bohonos, N.; Ullstrup, A.J. Decumbin, a new compound from a species of Penicillium. Nature 1958, 181, 1072−1073. https://doi.org/10.1038/1811072a0
  8. Renault, L.; Guibert, B.; Cherfils, J. Structural snapshots of the mechanism and inhibition of a guanine nucleotide exchange factor. Nature 2003, 426, 525−530. https://doi.org/10.1038/nature02197
  9. Lu, X.X.; Jiang, Y.Y.; Wu, Y.W.; Chen, G.Y.; Shao, C.L.; Gu, Y.C.; Liu, M.; Wei, M.Y. Semi-Synthesis, Cytotoxic Evaluation, and Structure—Activity Relationships of Brefeldin A Derivatives with Antileukemia Activity. Drugs 2022, 20, 26. https://doi.org/10.3390/md20010026
  10. Prieto-Dominguez, N.; Parnell, C.; Teng, Y. Drugging the small GTPase pathways in cancer treatment: Promises and challenges. Cells 2019, 8, 255–280. https://doi.org/10.3390/cells8030255
  11. Anadu, N.O.; Davisson, V.J.; Cushman, M. Synthesis and anticancer activity of Brefeldin A ester derivatives. Med. Chem. 2006, 49, 3897–3905. https://doi.org/10.1021/jm0602817

Reviewer 2 Report

This manuscript reports the systematic study of endophytic fungus from the mangrove plant, Acanthus ilicifolius. Fungi were isolated from the plant, and the diversity of mangrove fungi was explored. Four genera Verticillium, Neocosmospora, Valsa, and Pyrenochaeta were firstly isolated from mangrove trees. Thirty-one strains of fungi displayed strong cytotoxicity to different cell lines, and the genus Penicillium was found as the anticancer-producing strain. This work showed that combining activity-guided strategy with fingerprint analysis may improve the efficiency of discovering active compounds. The compounds responsible for anticancer activity was identified, for example, brefeldin A. This manuscript is recommended publication after minor revision. In order to improve this manuscript, please consider the comments and suggestions, which are listed below.
1. In fact, mangrove associated fungi are endophytic fungi. So, the keywords should include the word “endophytic fungi”.
2. Keywords should include the word “Fungal metabolites”.
3. “The endophic fungi derived from A. ilicifolius”; misspelling for “endophic”. Please correct.
4. Please revise “the target active strains were quickly locked” to “the target active strains were quickly identified”.
5. “The fungal community contains comprising Penicillium (21 isolates),”; please re-write and check this sentence.
6. Please re-write these phrases “Take appropriate samples and use scissors or scalpel to cut all parts, including root, stem and leaf. Pay attention to the integrity of sampling.”. They should be sentences. Please also correct such issue throughout the manuscript, please see, for example, at “Observe whether a single fungal colony grows and then transerred the fungi onto new PDA plates on the basis of their morphological differences.”. This is not a sentence.
7. This work found that fungi of the genus Penicillium have unique anticancer property. Please discuss previous reports on potent anticancer metabolites from the genus Penicillium, please see, for example, Cytotoxic metabolites from the endophytic fungus Penicillium chermesinum: discovery of a cysteine-targeted Michael acceptor as a pharmacophore for fragment-based drug discovery, bioconjugation and click reactions, RSC Adv., 2015,5, 70595-70603

Author Response

Reviewer #2

This manuscript reports the systematic study of endophytic fungus from the mangrove plant, Acanthus ilicifolius. Fungi were isolated from the plant, and the diversity of mangrove fungi was explored. Four genera Verticillium, Neocosmospora, Valsa, and Pyrenochaeta were firstly isolated from mangrove trees. Thirty-one strains of fungi displayed strong cytotoxicity to different cell lines, and the genus Penicillium was found as the anticancer-producing strain. This work showed that combining activity-guided strategy with fingerprint analysis may improve the efficiency of discovering active compounds. The compounds responsible for anticancer activity was identified, for example, brefeldin A. This manuscript is recommended publication after minor revision. In order to improve this manuscript, please consider the comments and suggestions, which are listed below.

  1. In fact, mangrove associated fungi are endophytic fungi. So, the keywords should include the word “endophytic fungi”. 2. Keywords should include the word “Fungal metabolites”. 3.“The endophic fungi derived from A. ilicifolius”; misspelling for “endophic”. Please correct. 4. Please revise “the target active strains were quickly locked” to “the target active strains were quickly identified”. 5. “The fungal community contains comprising Penicillium (21 isolates),”; please re-write and check this sentence. 6. Please re-write these phrases “Take appropriate samples and use scissors or scalpel to cut all parts, including root, stem and leaf. Pay attention to the integrity of sampling.”. They should be sentences. Please also correct such issue throughout the manuscript, please see, for example, at “Observe whether a single fungal colony grows and then transerred the fungi onto new PDA plates on the basis of their morphological differences.”. This is not a sentence. 7. This work found that fungi of the genus Penicillium have unique anticancer property. Please discuss previous reports on potent anticancer metabolites from the genus Penicillium, please see, for example, Cytotoxic metabolites from the endophytic fungus Penicillium chermesinum: discovery of a cysteine-targeted Michael acceptor as a pharmacophore for fragment-based drug discovery, bioconjugation and click reactions. RSC Adv., 2015,5, 70595-70603.

Response:

Thanks for your suggestion. We have added the word “endophytic fungi” and “Fungal metabolites” in the keywords. The sentence “The endophic fungi derived from A. ilicifolius” has been revised as “endophytic”. We have revised the sentence as “the target active strains were quickly identified” and rewritten the sentence “The fungal community contains comprising Penicillium (21 isolates),” as “The fungal community was dominated by Penicillium, comprising 21 isolates,”. We have rewritten the sentence “Observe whether a single fungal colony grows and then transferred the fungi onto new PDA plates on the basis of their morphological differences” as “Re-plated the fungi onto new PDA plates several times until the morphology of the fungi could be distinguished.” in our manuscript. We have added the previous reports on potent anticancer metabolites from the genus Penicillium, such as cytotoxic metabolites were isolated from the endophytic fungus Penicillium chermesinum: discovery of a cysteine-targeted Michael acceptor as a pharmacophore for fragment-based drug discovery, bioconjugation and click reactions [25]. The heteroatom-containing new compounds 2-hydroxyl-3-pyrenocine-thio propanoic acid and 5,5-dichloro-1-(3,5-dimethoxyphenyl)-1,4-dihydroxypentan-2-one, which isolated from a deep-sea Penicillum citreonigrum XT20-134 showed potent cytotoxicity to the human hepatoma tumor cell Bel7402 [26].

  1. Darsih, C.; Prachyawarakorn, V.; Wiyakrutta, S.; Mahidol, C.; Ruchirawat, S.; Kittakoop, P. Cytotoxic metabolites from the endophytic fungus Penicillium chermesinum: discovery of a cysteine-​targeted Michael acceptor as a pharmacophore for fragment-​based drug discovery, bioconjugation and click reactions. RSC Adv., 2015, 5, 70595–70603. https://doi.org/10.1039/C5RA13735G
  2. Tang, X.X.; Liu, S.Z.; Yan, X.; Tang, B.W.; Fang, M.J.; Wang, X.M.; Wu, Z.; Qiu, Y.K. Two New Cytotoxic Compounds from a Deep-Sea Penicillum citreonigrum XT20-134. Mar. Drugs 2019, 17, 509; doi:10.3390/md17090509

Round 2

Reviewer 1 Report

Dear authors, you wrote the introduction very well and clearly. 

In general, we can assume that the authors answered all the questions and made the appropriate changes.